# Genome Integrity and Neurological Disease

**DOI:** 10.3390/ijms23084142

**Published:** 2022-04-08

**Authors:** Elle E. M. Scheijen, David M. Wilson

**Affiliations:** Biomedical Research Institute, Hasselt University, 3590 Diepenbeek, Belgium; elle.scheijen@uhasselt.be

**Keywords:** oxidative DNA damage, DNA repair, neuronal cell function, neurodegeneration, inherited disorders, proliferation status

## Abstract

Neurological complications directly impact the lives of hundreds of millions of people worldwide. While the precise molecular mechanisms that underlie neuronal cell loss remain under debate, evidence indicates that the accumulation of genomic DNA damage and consequent cellular responses can promote apoptosis and neurodegenerative disease. This idea is supported by the fact that individuals who harbor pathogenic mutations in DNA damage response genes experience profound neuropathological manifestations. The review article here provides a general overview of the nervous system, the threats to DNA stability, and the mechanisms that protect genomic integrity while highlighting the connections of DNA repair defects to neurological disease. The information presented should serve as a prelude to the Special Issue “Genome Stability and Neurological Disease”, where experts discuss the role of DNA repair in preserving central nervous system function in greater depth.

## 1. Nervous Systems and the Brain

Nervous systems within the animal kingdom vary tremendously in structure and complexity [1,2]. Simple nervous systems can range from the non-centralized neural network in jellyfish to sea sponges who lack a true nervous system altogether. Unlike most invertebrate species, the nervous systems of vertebrates are complex, centralized, and specialized. While there is diversity among vertebrates, they share a fundamental nervous system structure, i.e., a central nervous system (CNS) that consists of a brain and spinal cord and a peripheral nervous system (PNS) composed of peripheral sensory and motor nerves. As a whole, the nervous system works coordinately to receive and transmit complex information throughout the body, directing all aspects of thought, behavior, and physical activity.

The brain serves as the master receiver, organizer, and distributor of sensory and visual information from the body, controlling our thoughts, memory and speech, physical movements, and the function of most organs [3,4]. Neurons are the basic functional units of the brain, producing electrical pulses to transmit information across great distances within the interconnected circuitry of the body. Glia cells, i.e., astrocytes, oligodendrocytes, and microglia, primarily operate in the mature CNS to support neurons [5], with data indicating unique and specialized roles for neuroglia in preserving the proper operation of the nervous system. Astrocytes function to maintain brain homeostasis and neuronal metabolism by producing antioxidants, recycling neurotransmitters, and supporting the blood–brain barrier. Oligodendrocytes are responsible for generating the insulating myelin sheath around axons, enabling easy and efficient transfer of electrical signals between nerve cells. Microglia are specialized macrophages and operate to remove damaged neurons or infections to safeguard brain health. Disruption of any one of these components can lead to biochemical and cellular abnormalities that give rise to CNS dysfunction and disease.

## 2. Neurodegenerative Diseases

The National Institute of Neurological Disorders and Stroke within the United States of America lists over 400 distinct neurological diseases on its website, and it is almost certainly not exhaustive (https://www.ninds.nih.gov/disorders/all-disorders, accessed on 2 March 2022). These include neurogenetic diseases, developmental disorders, degenerative diseases, and trauma or infection-induced diseases. Neurodegenerative diseases, in particular, involve the progressive loss or death of nerve cells and directly affect the lives of hundreds of millions of people worldwide, encompassing diseases such as Alzheimer disease (AD), Parkinson disease (PD), Huntington disease (HD), and amyotrophic lateral sclerosis (ALS). The symptomatic degeneration and accompanying brain atrophy present in these disorders result in impaired memory or thought processes, confusion and anxiety, mood swings, poor coordination, and/or loss of movement control. While the causes of neurodegenerative diseases are very heterogeneous in nature and are still being deciphered, the most common risk factor is age, a phenomenon known to involve DNA damage and genomic instability, epigenetic alterations, loss of proteostasis, mitochondrial dysfunction, and altered intercellular communication, to name a few of the hallmarks [6]. In combination with the numerous basic research and clinical investigations conducted to date, current evidence, therefore, implicates the above molecular mechanisms, as well as oxidative stress, protein aggregation, cytoskeletal abnormalities, synaptic dysfunction, and cell death pathways as underlying drivers of neurodegenerative disease [7]. This Special Issue is dedicated to more exhaustively enumerating the link between DNA damage, genomic stress, neuronal cell loss, and neurological disease.

## 3. Exogenous and Endogenous Threats to the Brain: Oxidative DNA Damage

The compositional integrity of organisms is constantly challenged by a barrage of insults from both external (exogenous) and internal (endogenous) sources, which often possess the ability to generate lipid, protein, RNA, and/or DNA damage [8]. Some of the most common external sources include sunlight, ionizing radiation (IR), and chemicals or pollutants found in the environment. In addition, many agents used as clinical therapeutics have either intended or incidental harmful effects on macromolecule integrity [9]. Given that damage to the first three cellular components (i.e., lipids, proteins, and RNA) can be resolved by degradation and molecular turnover, alterations to the integrity of the genetic blueprint (DNA) are thought to be the most threatening to organism function and health. Beyond the array of external chemical and physical agent exposures, exogenous stress can also include physical trauma (e.g., accidents), lifestyle choices (e.g., drug or alcohol use), or unexpected changes in one’s personal life or mental health (e.g., due to the loss of a loved one, work problems, etc.) [10]. Collectively, the impact of external stress can be significant, leading to deleterious changes in the composition of macromolecular structures and the effectiveness of biochemical processes.

In addition to the numerous external threats, several endogenous agents can cause macromolecular damage [11], including to DNA. Perhaps the best known intracellular damaging agents are reactive oxygen species (ROS), formed as natural by-products during mitochondrial respiration in the process of generating energy (ATP). ROS can create a range of base and sugar modifications upon reacting with DNA, some of which promote strand breakage. Through reactions with phospholipids, ROS can generate epoxide or aldehyde species, which can, in turn, react with DNA to produce severe genomic lesions, such as interstrand crosslinks that covalently connect the two strands of the helix [12]. Apart from ROS-induced DNA damage, S-adenosylmethionine, a common co-substrate involved in methyl group transfers, including during epigenetic landscaping, has been found to react with DNA, producing alkylated base products such as N3-methyladenine and O6-methylguanine [13]. Lastly, DNA, as a chemical, exhibits intrinsic instability, mainly in the form of deamination of cytosine to uracil or the loss of purine or pyrimidine bases, creating an abasic site that lacks instructional coding information. Collectively, it has been estimated that a typical human genome will experience over 100,000 alterations per day under normal physiological conditions [8].

ROS are not only continuously being produced and leaked by mitochondria but are also commonly induced by exogenous stressors, such as those described above. Indeed, IR is well known for generating ROS through the radiolysis of water [14], a fact that seeded the free-radical theory of aging proposed by Denham Harman in 1952 [15]. He argued that aging and progressive degenerative disease occur due to the gradual accumulation of harmful oxidative damage during the lifetime of an organism. Although the free radical theory has encountered setbacks, the preponderance of evidence supports the notion that oxidative macromolecular damage accumulates with age [16]. Since the discovery of the intrinsic instability of DNA and the role of genomic stress in disease, many researchers have pursued the idea that the DNA template is the primary pathological target of ROS.

It is important to emphasize that, unlike the PNS, the CNS is protected from most exogenous threats by the vertebral column that encircles the spinal cord and the skull that houses the brain [17]. In addition, the CNS is shielded from many circulating external factors, such as toxins, by the blood–brain barrier [18]. Thus, unlike the PNS, which appears to be more susceptible to the deleterious effects of external damaging agents, the CNS is mainly challenged by endogenous stressors. The brain is one of the most active and energetically demanding organs in the human body, consuming roughly 49 mL of oxygen per minute, with nearly all oxygen being utilized for the oxidation of carbohydrates (almost exclusively glucose) to produce high-energy ATP [19]. As oxidative phosphorylation via the electron transport chain in mitochondria is the primary source of ATP generation, the risk of ROS production is high in neuronal cells [20]. The primary front-line defense against ROS is comprised of a battery of scavenging molecules and enzymes, collectively known as antioxidants [21]. Thus, how much oxidative damage is generated depends on the balance between ROS production and neutralization by antioxidants. Since the brain has been reported to maintain a comparatively low level of antioxidant systems, there is an increased need for efficient oxidative damage coping mechanisms, including DNA repair processes that resolve ROS-induced DNA damage and preserve genome integrity [22].

## 4. DNA Repair Mechanisms

Since evidence implicates oxidative stress and associated macromolecular damage as a critical element in the etiology of neurodegenerative disease, we focus the remainder of the review on DNA damage and DNA repair. Oxidative DNA damage includes a range of alterations, such as base modifications (e.g., 8-oxoguanine or cyclopurines), protein-DNA adducts, DNA-DNA crosslinks, abasic sites, strand breaks, and clustered lesions [23,24]. Depending on the chemical, structural, and positional nature, DNA damage can promote mutagenesis, genomic instability, or stress responses triggered upon replication or transcription arrest [25] (Figure 1). Considering the non-dividing nature of terminally differentiated neurons, it is likely the loss of genome integrity or the persistent stalling of RNA polymerase II at blocking damage that activates a cell death response, giving rise to neurodegeneration. Additionally, unresolved DNA damage may promote cell cycle activation and reentry via a DNA damage response mechanism, leading to apoptotic cell death as well [26].

As a means of understanding the fundamental contributions of the different DNA repair pathways (and sub-pathways), we provide next an overview of the major corrective mechanisms in humans: (1) direct reversal, (2) mismatch repair (MMR), (3) base excision repair (BER), (4) single-strand break repair (SSBR), (5) ribonucleotide excision repair (RER), (6) nucleotide excision repair (NER), and (7) double-strand break repair (DSBR). In general, excluding direct reversal (see more below), DNA repair involves some variation of the following biochemical steps: (a) damage recognition; (b) damage processing, excision, or resolution; (c) where necessary, replacement of the removed genetic material; and (d) strand break closure. We describe the *nuclear* DNA repair mechanisms below, yet it is important to recognize that separate, albeit often related, corrective pathways exist within the *mitochondrial* compartment to deal with mitochondrial DNA (mtDNA) damage [27]. Table 1 is included as a reference of the DNA repair pathways, the central protein components and main biochemical activities, and the connections to inherited (neurodegenerative) disorders.

### 4.1. Direct Reversal

In direct reversal, the biochemical event entails the precise removal of subtle base modifications from DNA, immediately restoring the genome to its original, undamaged state (Figure 2). For instance, it was discovered early during DNA repair investigations that organisms possess an evolutionarily conserved mechanism for resolving O^6^-methylguanine adducts. The human protein, termed O^6^-methylguanine DNA methyltransferase (MGMT), like its bacterial functional counterparts (i.e., *Escherichia coli* Ada and Ogt), utilizes a conserved active site cysteine residue to directly remove alkylation damage from DNA [28,29,30]. This repair step results in the permanent inactivation of the protein and its consequent degradation, commonly referred to as a “suicide” mission [31].

In addition, recent investigations have uncovered nine human proteins (ALKBH1-8 and FTO), sharing homology with the bacterial DNA repair enzyme ALKB as part of the family of Fe(II)- and α-ketoglutarate-dependent dioxygenases [32,33]. Of the nine paralogs, ALKBH2 and ALKBH3 are bonafide nuclear DNA repair enzymes, possessing the ability to directly remove 1-methyladenine, 1-ethyladenine, 3-methylcytosine, and specific etheno adducts from damaged polynucleotides, sometimes with single-stranded specificity, in an ɑ-ketoglutarate-dependent reaction [34,35,36]. ALKBH1 is also a nucleic acid demethylase with the ability to target both RNA and DNA substrates but localizes to the mitochondria [37]. Most of the other ALKB homologs demethylate RNA or proteins, functioning in gene expression regulation or epigenetic programming, or have biochemical activities that have yet to be defined.

### 4.2. Mismatch Repair

To cope with mispaired nucleotides or small insertions/deletions that arise as DNA replication errors, organisms have evolved the MMR system [38] (Figure 3). MMR, which is likely intimately connected to the replication machinery to ensure faithful removal of the newly synthesized material, entails a scanning and recognition step, followed by the excision of the portion of the daughter strand that contains the incorrectly inserted nucleotide or bulge/loop structure. A complex of MutS homologs carries out damage detection, i.e., either MSH2/MSH6 (MUTSɑ, for base–base mismatches and small loops) or MSH2/MSH3 (MUTSβ, for small and large loops) [39,40]. Subsequent excision of the misincorporated nucleotides is mediated by exonuclease 1 (EXO1) or the MUTLɑ complex (MLH1 and endonuclease PMS2), in coordination with the MUTS recognition complex, replication protein A (RPA), replication factor C (RFC), proliferating cellular nuclear antigen (PCNA), and DNA polymerase δ (POLδ) [41,42,43,44]. Following damage excision, the remaining steps include re-replication of the now missing section of DNA (likely by the replicative polymerase POLδ) and closure of the remaining nick (likely by ligase 1 (LIG1) in cooperation with PCNA).

### 4.3. Ribonucleotide Excision Repair

Driven by the observation that as many as 1 million or more ribonucleotides can be incorporated during a single round of chromosome replication, recent attention has been put into deciphering the molecular steps of RER [45,46] (Figure 3). In short, RER looks a lot like long-patch BER (see next), except that the initial incision event is catalyzed by RNaseH2, a heterotrimeric complex that also degrades RNA:DNA hybrids [47]. After strand cleavage 5′ of the faulty ribonucleotide, the replicative polymerases (POLδ/POLε) carry out strand displacement synthesis with RFC and PCNA, followed by removal of the 5′-flap containing the damage via flap endonuclease 1 (FEN1) and nick ligation by LIG1 [48]. There appears to be an alternative topoisomerase 1 (TOPO1)-mediated processing mechanism, which we will not discuss here.

### 4.4. Base Excision Repair

BER is the primary pathway for coping with spontaneous decay products (e.g., apurinic/apyrimidinic (AP) sites for base loss or uracil from cytosine deamination) and oxidative base or sugar lesions [49] (Figure 4). Classic BER begins with the removal of a substrate base by a DNA glycosylase, resulting in the generation of an AP site intermediate. In humans, there are 11 known DNA glycosylases: uracil DNA glycosylase (UNG), methylpurine DNA glycosylase (MPG), nth endonuclease III-like (NTHL1), 8-oxoguanine DNA glycosylase (OGG1), mutY homology (MUTYH), thymine DNA glycosylase (TDG), single-strand-selective monofunctional uracil DNA glycosylase (SMUG1), methyl CpG binding domain protein 4 (MBD4), and the nei endonuclease VIII-like proteins (NEIL1, NEIL2, and NEIL3) [50]. The AP site intermediate is then primarily incised by APE1, creating an SSB with a priming 3′-hydroxyl group and a damaged 5′-abasic fragment [51]. The missing nucleotide is then replaced, and the 5′-end processed to create a regular 5′-phosphate group to permit the ligation and completion of the repair response. The primary gap-filling polymerase in humans is POLβ, which also possesses the ability to remove 5′-abasic groups (i.e., the deoxyribose phosphate) left behind by APE1 incision [52]. X-ray cross-complementing protein 1 (XRCC1) in complex with DNA LIG3ɑ plays a role in sealing the nick during classic BER [53,54]. Besides the single-nucleotide (a.k.a., short-patch) BER event described above, humans possess a long-patch BER process that entails strand-displacement synthesis, typically by a replicative polymerase (POLδ/POLε) in collaboration with PCNA, 5′-flap resolution by FEN1, and nick sealing by LIG1 [55].

### 4.5. Single-Strand Break Repair

Related to BER, yet more appropriately considered an independent pathway, SSBR has been designed to deal with strand breaks that harbor a non-conventional terminus created as an intermediate during DNA enzymatic processing or through direct chemical damage to DNA [56] (Figure 4). The most prominent of the damaged termini include: 3′-phosphate, a product of ROS attack of the sugar moiety in DNA and certain enzymatic events; 3′-phosphoglycolate and 5′-aldehyde, common to IR exposure and also products of ROS attack; 3′-phospho-ɑ,β-unsaturated aldehyde, an intermediate of a dual functional DNA glycosylase with AP lyase activity; 3′-TOPO1 adduct, a trapped enzyme intermediate due to the presence of interfering DNA damage or exposure to a therapeutic agent like camptothecin; 5′-hydroxyl group, a product of IR exposure and certain alkylators; 5′-adenylate (AMP), an intermediate of failed DNA ligation; and the 5′-deoxyribose phosphate that arises following APE1 incision during BER. These different lesion types must be resolved to create normal 3′-hydroxyl and 5′-phosphate ends to permit repair synthesis and/or nick ligation [57].

SSB lesions are often initially recognized by poly(ADP)ribose polymerase 1 (PARP1), a strand break sensor that operates to coordinate the repair response through DNA binding-mediated post-translational modifications [58]. Then, depending on the type of termini present at the strand break ends, different enzymes are needed to facilitate the SSBR event. APE1, the major AP endonuclease in BER, also plays a significant role in SSBR, specifically processing 3′-phosphoglycolates and 3′-α,β-unsaturated aldehydes and to a lesser extent 3′-phosphate groups [59,60]. Aprataxin (APTX) copes with 3′-phosphoglycolates but maintains a more critical role in resolving 5′-adenylate intermediates [61]. Polynucleotide kinase 3′-phosphatase (PNKP), as the name suggests, removes 3′-phosphate blocking groups and phosphorylates 5′-hydroxyl termini [62,63]. Tyrosyl-DNA phosphodiesterase 1 (TDP1) has evolved to remove 3′-TOPO1 adducts, which are likely processed to small peptides before being converted to 3′-phosphate and eventually 3′-hydroxyl ends [64]. FEN1, the predominant 5′-flap endonuclease, as well as other structure-specific nucleases, such as the complex consisting of excision repair cross-complementing 1 and xeroderma pigmentosum complementation group F (i.e., ERCC1/XPF), also appear to play a general role in termini clean-up [65,66,67]. FEN1, for instance, assists in the removal of 5′-aldehyde and 5′-adenylate groups after strand displacement repair synthesis events [68,69]. POLβ excises 5′-deoxyribose abasic fragments that arise during classic BER and appears to also contribute to 5′-adenylate removal via its associated lyase activity. After termini resolution and gap-filling, SSBR, like BER, is typically completed by either the XRCC1/LIG3ɑ complex or LIG1 [70].

### 4.6. Nucleotide Excision Repair

NER is divided into two major sub-pathways, global-genome (GG-NER) and transcription-coupled (TC-NER), molecular processes that cope with large, helix-distorting adducts, most notably UV-light induced (6-4)photoproducts and cyclopyrimidine dimers [71,72] (Figure 5). The distinction between the two pathways is centered around their damage recognition mechanism. In the case of GG-NER, which, as the name implies, handles substrates throughout the genome, damage identification is mediated by rad23 homolog B (RAD23B) and the XPC protein, which recognize disrupted base pairs within the duplex; damage verification is performed in concert with RPA, XPA, and XPD, a helicase within the transcription complex TFIIH [73,74]. For TC-NER, the substrate lesion, when positioned on the transcribed strand, is revealed by the stalling of RNA polymerase during transcription elongation [75]. This event calls into action numerous factors, including the TFIIH complex, XPA, RPA, and the TC-NER proteins, Cockayne syndrome A (CSA, a.k.a., ERCC8), and CSB (a.k.a., ERCC6) [76,77]. Following the recognition and verification of the damage, the two NER sub-pathways involve identical proteins that carry out excision of the damage-containing strand (ERCC1/XPF, 5′ incision; XPG, 3′ incision), replacement of the missing DNA segment (POLδ/POLε), and ligation of the remaining nick (LIG1 in replicating or XRCC1-LIG3α complex in non-replicating cells) [78].

### 4.7. Double-Strand Break Repair

Cells have adopted two primary corrective responses to cope with DSBs: homologous recombination (HR) and non-homologous end-joining (NHEJ) (Figure 6). DSBs exist in two primary forms, one-ended and two-ended. The former is an intermediate of replicative stress, arising after fork collapse at sites of DNA damage (e.g., SSBs), complex sequences (e.g., microsatellites), or active RNA synthesis and R-loop formation. The latter is created upon the breakage of both strands of DNA, either via chemical attack or enzymatic processing. Beyond just the cell cycle phase, several factors dictate pathway choice, i.e., HR or NHEJ [79,80]. We focus here on only the key molecular features of the two processes.

HR, which also operates to introduce genetic diversity into the gene pool during meiosis, faithfully resolves both one-ended and two-ended DSBs using a homologous sequence (typically the sister chromatid) as a template, thereby restricting the process to primarily the S/G2 phases of the cell cycle [81,82]. In brief, HR entails the following coordinated steps: (i) damage recognition, signaling, and some initial processing by the MRN complex (composed of meiotic recombination 11 homolog 1 (MRE11), RAD50, and Nijmegen breakage syndrome 1 (NBS1)); ataxia telangiectasia mutated (ATM); and RB binding protein 8 (RBBP8; a.k.a., CtIP and SAE2), (ii) long-range resection of the DSB end to create the necessary RPA-coated 3′-single-stranded tail for strand invasion (by exonuclease 1 (EXO1) and DNA2, likely in coordination with the Bloom helicase (BLM)), (iii) strand exchange into the undamaged sister chromatid and D-loop formation via RAD51 filament formation and the activities of BRCA2, the RAD51 paralogs, RAD54, among possible others, (iv) transfer of genetic information from the sister chromatid to the damaged chromosome through orchestrated DNA movements (e.g., branch migration) and DNA synthesis most likely by POLδ, and (v) resolution of the intertwined duplexes, the so-called Holliday junction (by BLM and TOPO3ɑ or a nuclease complex, such as MUS81/EME1 (essential meiotic structure-specific endonuclease 1)). There are alternative forms of HR and different mechanisms of proceeding through DSB resolution that we will not cover here (see references above and [83]). Still, in the end, the process typically results in faithful removal of DSBs from the genome.

NHEJ, which likely evolved to execute programmed processes, such as class switch recombination that facilitates antibody diversification, operates independently of the cell cycle yet is most prominent in G1 [84]. The classic pathway entails the primary steps [85]: initial recognition of the DSB by the Ku complex, consisting of KU70 and KU80, and DNA-PK (catalytic subunit); alignment of the two ends of the two-ended DSB; and end-processing by MRN, Artemis, other clean-up enzymes (see SSBR), or DNA polymerases (e.g., POLλ and POLμ). These biochemical steps facilitate annealing and ligation by the XRCC4-like factor (XLF), a paralogue of XRCC4 and XLF (PAXX), XRCC4, and LIG4. NHEJ tends to be error-prone as small segments of DNA are often added or removed during the end-joining step. Recent work has identified two sub-pathways of NHEJ; classic and alternative (a.k.a., microhomology-mediated end-joining), with the latter involving a distinct set of proteins (e.g., PARP1, XRCC1, FEN1, MRE11, NBS1, LIG3, and POLθ), more commonly inducing extensive genomic alterations (typically deletions), and often being upregulated in cancer as a compensatory system [86].

## 5. DNA Repair Defects in Neurodegenerative Disease

Armed with the knowledge of the previous section, we next review the evidence for the involvement of DNA damage and DNA repair mechanisms in neurodegenerative disease. We focus initially on inherited disorders that experience neurodegeneration and subsequently on associations of DNA repair defects with broader neurodegenerative disorders.

### 5.1. Inherited Disorders Involving Defects in DNA Repair and Neurological Abnormalities

The suspected participation of DNA damage in neurodegeneration was validated upon the discovery that inherited disorders involving neurological abnormalities, such as xeroderma pigmentosum (XP), stem from defects in the ability to efficiently respond to and clear genomic stress [87]. XP, characterized by extreme sun sensitivity and a high occurrence of ultraviolet radiation-induced skin cancers, was the first human genetic disorder to be shown to harbor a genuine DNA repair defect [88,89]. XP is comprised of complementation groups, XPA–XPG, which identify factors of the NER sub-pathways that have specific roles in removing obstructive, helix-distorting DNA adducts, such as ultraviolet irradiation photoproducts. Besides being cancer-prone, approximately a quarter of XP-affected individuals exhibit progressive neurological degeneration. Since mutations in genes that affect only GG-NER (e.g., XPC) result in strictly cancer predisposition, whereas mutations in TC-NER-associated genes (i.e., CSA, CSB, XPA, XPB, XPD, XPF, and XPG [90]) give rise to neurological defects, it is believed that persistent transcription blocking lesions are particularly lethal to neuronal cells. The primary role for NER in the brain is presumed to be the removal of certain forms of oxidative DNA damage, such as 8-oxoguanine, apurinic/apyrimidinic (AP or abasic) sites, cyclopurines, DNA-DNA crosslinks, and protein-DNA adducts, particularly since in many instances alternative resolution mechanisms are absent, namely replication-associated HR [91,92].

Neurological problems occur not only due to deficits in TC-NER but also due to defects in other DNA repair processes [93,94]. Ataxia telangiectasia (AT) is a rare, monogenic recessive disorder characterized by radiosensitivity, increased cancer risk (particularly lymphomas), reduced immunological function, and progressive neurodegeneration that gives rise to ataxia, chorea, myoclonus, and neuropathy [95]. The defective gene in AT, i.e., AT mutated (ATM), encodes a serine/threonine-protein kinase that functions as a central signaling enzyme in the response to DSBs [96,97]. Its role in the faithful resolution of recombinogenic DNA DSBs explains the hypersensitivity of AT cells to IR, a physical agent that creates DSBs, and the pronounced genomic instability. However, given that homology-direct repair is not fully operational in non-dividing cells due to the lack of a paired sister chromatid, the precise role of ATM in protecting against progressive neuronal degeneration is less clear [98]. Supportive of a role for DNA DSBs in the neurodegenerative features of AT, inherited mutations in the nuclease meiotic recombination 11 homolog (MRE11), a protein that collaborates with ATM in the recognition and response to DNA DSBs, results in AT-like disorder (ATLD), a disease that exhibits similar neurological defects as AT [99]. While in these two diseases, some of the pathologies might arise from problems during development when cells are rapidly dividing, the absence of microcephaly in AT or ATLD individuals points towards brain atrophy mainly originating later in life [100]. Recent studies have indicated that DNA DSBs may arise in non-replicating, mature neurons through the action of ROS or the cleavage activity of TOPOIIꞵ as part of normal gene expression regulation [101]. However, the precise resolution mechanism(s) and how it might engage ATM remains unclear. Evidence also implicates ATM in mitigating cellular oxidative homeostasis, possibly through cytoplasmic or mitochondrial functions, and a nuclear DNA repair response to less complex oxidative DNA damage, such as SSBs, particularly in the context of transcription [98,102].

Besides ATM and MRE11, defects in other proteins that operate to clear DSBs or replicative stress have been tied to distinct disorders that involve neurological disease, cancer predisposition, and a range of other clinical phenotypes: Nijmegen breakage syndrome 1 (NBS1), ataxia telangiectasia and Rad3-related (ATR), DNA ligase 4 (LIG4), Cernunnos (a.k.a., XLF or NHEJ1), and components of the Fanconi anemia pathway (Table 1). However, unlike AT, these disorders mostly exhibit microcephaly, likely reflective of failed cell duplication during embryogenesis and consequent poor brain development [100].

Defects in components of SSBR (e.g., APTX, TDP1, PNKP, and XRCC1) have also been associated with genetic disorders that present with neurological disease (namely ataxia) [56]; yet these syndromes exhibit no overt cancer predisposition or other shared clinical manifestations with the diseases mentioned above (Table 1). The restrictive nature of the clinical presentation in SSBR diseases to neurological deficits presumably stems from the fact that elevated endogenous strand breaks in non-dividing neurons result in transcription arrest and the activation of cell death pathways. Conversely, such SSB intermediates are resolved faithfully by replication-directed HR in dividing cells. Additionally, recessive mutations in Senataxin (SETX), an RNA-DNA helicase, have been linked to apraxia with oculomotor ataxia type 2 (AOA2), whereas dominant mutations in the gene have been detected in a juvenile-onset form of ALS [103]. Amassing evidence indeed suggests that defects in R-loop resolution, i.e., the ability to disentangle RNA-DNA hybrids, which are frequent intermediates of transcription, result in loss of RNA transcriptome homeostasis and genome integrity, culminating in neuronal cell death [104,105]. Lastly, the accumulation of nuclear DNA strand breaks can result in the hyperactivation of PARP1, leading to NAD overconsumption and mitochondrial dysfunction that contribute to the neurodegenerative process [106].

### 5.2. Associations of DNA Repair Defects with Classic Neurological Diseases

Besides the direct link between mutations in established DNA repair genes and the inherited disorders mentioned above (Table 1), DNA damage accumulation or defects in DNA damage processing have been implicated in nearly all neurological diseases, including sporadic and familial cases of AD, ALS, and PD [107,108,109]. In these latter diseases, oxidative DNA damage, which often arises in concert with mitochondrial dysfunction, is typically higher in disease tissue than in comparative control tissue. Evidence indicates that defects in BER, the primary system for handling oxidative DNA damage, can promote neurological degeneration in AD models, implying at minimum a protective role for DNA repair in disease etiology [110]. In addition, AD is associated with the accumulation of DNA DSBs [111], particularly in vulnerable neuronal and glial cell populations, likely due to reduced expression of DSB response proteins (e.g., MRE11, RAD50, and BRCA1) [112]. Likewise, in ALS, mutations in the superoxide dismutase 1 (SOD1) gene lead not only to defective ROS scavenging and oxidative homeostasis but to the impaired control of important DNA damage response gene expression (e.g., of SPY1) [113]. Moreover, mutations in TARDBP, which account for ~4% of familial ALS cases, result in the pathogenic mislocalization of the TDP43 protein from the nucleus to the cytoplasm, causing defects in cytoplasmic-nuclear trafficking and reduced levels of proteins in the nucleus, such as the NHEJ factor XRCC4/LIG4 [114]. FUS, another protein found to be defective in sporadic and familial forms of ALS, has been demonstrated to have more direct roles in DNA repair, recruiting XRCC1/LIG3 to SSBs and promoting strand break resolution, as well as modulating DSBR efficiency, possibly through interactions with the chromatin modifier HDAC1 [115,116,117]. Mutations in FUS lead to cytoplasmic aggregation and loss of normal protein function in the nucleus, culminating in genomic instability and neuronal cell death. Regarding PD, studies indicate that α-synuclein, a protein found to misfold and aggregate into the characteristic clumps called Lewy Bodies, activates the ATM kinase [118]. Elevated levels of poly(ADP)ribose, a symptom of persistent DNA damage, have also been reported to accelerate the fibrillization of α-synuclein [119]. Finally, similar to what has been observed in AD, compromised DNA repair, namely BER or NER, may serve as a risk factor for PD-like pathology [120,121].

Other proteins linked to inherited neurological disorders have also been described to regulate or participate in the response to DNA damage [122,123,124,125]: (i) the Huntingtin protein (defective in HD) is recruited to DNA damage sites by ATM and serves as a repair complex scaffold; (ii) ATNX3, defective in spinocerebellar ataxia 3 (SCA3), accumulates at sites of DNA damage, participates in the ATM response, and interacts with the SSBR protein PNKP and TC-NER complexes; and (iii) survival motor neuron (SMN1/SMN2) proteins, in which defects give rise to spinal motor atrophy (SMA) disease, associate with the HR protein RAD51 and influence the levels of the NHEJ factor DNA-PKcs and the RNA/DNA helicase SETX.

Given the lack of DNA replication in terminally differentiated cells, pathways such as MMR, which copes with replication errors, and classic HR, which resolves collapsed replication forks, are not thought to constitute major neuroprotective mechanisms. Seemingly consistent with this notion, defects in MMR give rise to microsatellite instability (MSI) and, almost exclusively, cancer susceptibility, with genetic links to a range of hereditary colorectal cancers. That said, it is worth noting that inherited defects have been identified in MMR components, the FANCD2/FANCI-associated nuclease 1 (FAN1), and LIG1 in individuals with HD, suggesting a role for an undefined DNA surveillance mechanism (presumably not classic MMR) in suppressing pathological CAG repeat expansion that defines HD [126]. Defects in factors that participate in DSB resolution and/or HR not only give rise to chromosome rearrangements, an outcome common to cancer and other pathological conditions associated with genomic instability, but also to immunological dysfunction and premature aging features, in addition to neurological disease.

### 5.3. DNA Damage, DNA Repair, and Neuronal Cell Function

The previous sections highlight the fact that defects in DNA repair can lead to genomic stress that is responsible for the neuronal cell deficiencies in neurodevelopmental or neurodegenerative diseases. However, emerging evidence is now pointing to a more nuanced role of DNA damage and repair in mitigating neuronal cell identity and function. Indeed, early work by Suberbielle et al. discovered that the natural behavior of young adult mice, i.e., exploration of a new environment, and increased neuronal activity causes the formation of DNA DSBs in neurons. This DNA strand breakage occurs at higher levels in the dentate gyrus, an area of the brain that plays a crucial role in learning, memory, and spatial coding [127]. With the advent of genome-wide DNA damage techniques, scientists now appreciate that DNA damage is in some cases generated in a region-specific manner with a physiological purpose. For instance, Madabhushi et al. reported that stimulation of neuronal activity triggers the formation of DNA DSBs in the promoters of a subset of early-response genes that are crucial for experience-driven changes to synapses, learning, and memory (i.e., Fos, Npas4, and Egr1), enabling the resolution of topological constraints and gene expression [128]. Additionally, Gadd45γ, a member of the growth arrest and DNA damage (Gadd45) protein family, operates after an initial wave of DNA DSB formation and DNA methylation in the prelimbic prefrontal cortex, directing a DNA repair-mediated DNA demethylation process that promotes temporal immediate early gene expression necessary for fear memory consolidation [129]. Studies have suggested functions for other DNA repair proteins, such as PARP-1 and FEN-1, in learning and memory formation as well [130,131]. Although the functional relevance remains unclear, more recent genome-wide analyses have revealed that post-mitotic neurons accumulate exceptionally high levels of DNA SSBs, as assessed by recurrent repair synthesis events, within specific genomic hotspots, namely enhancers at or near CpG dinucleotides and sites of DNA demethylation [132]. Thus, to maintain the integrity of neurons and their genome, DNA repair seems to preferentially protect essential genes within specific repair hotspots [133]. The above findings emphasize that DNA damage and repair may be targeted to specific genomic locations, possibly in unique ways in different cell types, brain regions, and biological contexts. Neuronal cell survival and functionality, therefore, may depend not only on the resolution of global DNA damage but on precise and efficient DNA repair processing in critical genomic domains. As such, DNA repair defects and associated DNA damage accumulation might not only lead to neurodegeneration directly but neuronal cell dysfunction more broadly, perhaps causing more widespread consequences on the CNS network.

## 6. Summary and Future Perspectives

DNA damage, and, as such, defective DNA repair, plays a causal role in numerous pathological states, including neurodegenerative phenotypes. The accumulation of DNA damage in neuronal cells either drives the loss of genome integrity or interferes with gene regulatory processes or transcriptional events, thereby promoting activation of cell death responses and consequent cell loss. As is likely evident from the presentation above, depending on the proliferative nature of the cell, a preference for certain DNA repair pathways exists, where for example, dividing cells are more reliant on systems such as MMR, RER, and HR, whereas non-dividing cells favor NHEJ, SSBR, or NER (Figure 7). Not surprisingly, given that neurons are non-dividing in nature, defects in DNA repair pathways such as SSBR and NER, which resolve obstructive DNA lesions largely independently of replicative status, have been linked to neurodegenerative outcomes. Conversely, pathways such as MMR, which predominantly cope with DNA replication errors, are not directly associated with neurological complications when impaired. Although there have been associations with BER defects and the development of neurological disease, BER gene mutations, which result in defective processing of oxidative DNA damage, are mostly restricted to cancer predisposition disorders without clear neurodegenerative features [134]. Nevertheless, as pointed out above, evidence that indicates a role for both BER and NER in regulating neurological disease risk and severity is mounting. It will be interesting going forward to determine specifically whether central BER defects play causal roles in neurodegenerative disease, particularly given the prominent role of the pathway in the repair of oxidative DNA damage.

As noted earlier, ribonucleotide incorporation into genomic DNA during replication is frequent. Of course, in non-dividing cells, faulty incorporation during chromosome duplication is not relevant. However, in situations of DNA repair synthesis, unwanted insertion of ribonucleotides could presumably still occur. Though it is presently unclear how this might affect the integrity of neuronal cell function, pathogenic mutations in RNAseH2, the main enzyme that initiates RER, give rise to Aicardi–Goutières syndrome (AGS), a monogenic type I interferonopathy characterized by neurodevelopmental defects and neuroinflammation [135]. Notably, recent evidence indicates that DNA damage-dependent signaling rather than type I interferon signaling underlies neurodegeneration in this class of disease [136]. While a prominent role for RER in the context of neurodevelopment is obvious, i.e., where replication is active, studies are needed to more precisely flush out the contribution of the repair system in terminally differentiated cells, such as neurons.

It is now well established that DNA DSBs can be generated in the neuronal genome through multiple mechanisms. While NHEJ is implicated in the resolution of such damage [128], the contributions of ATM and other HR-related factors are less clear [137]. Recent reports of an RNA-directed HR process involving RAD52 could provide non-dividing cells with the ability to carry out faithful repair of DSBs [138,139,140,141], as opposed to the error-prone mechanism of NHEJ that can give rise to chromosome instability. Delineating the molecular choreography that occurs at DSBs within the neuronal genome, both in terms of damage recognition and ultimately (ideally faithful) resolution, is a key topic of investigation looking into the future.

To date, studies have largely focused on the effects of DNA repair defects on neuronal cell function and viability. But as noted earlier, it is important to recognize that the nervous system is composed of an integrated network of neuronal and glial cell types. Determining the consequences of persistent DNA damage and the comparative role of the different DNA repair mechanisms in the distinct cell populations represents a critical step in defining the factors that influence health of the nervous system as a whole. Moreover, compared to the CNS, virtually nothing is known about DNA repair processes in the PNS or enteric nervous system, the other two major nervous systems in mammals. A broader understanding of DNA damage and repair in all nervous systems and within the cellular network of a particular nervous system will not only shed new light on the etiology of neurodegenerative diseases but can pave the way for new therapeutic strategies. Given the decades’ worth of research that has been poured into understanding the role of DNA repair in neurons within the CNS, it would appear that neuroscientists interested in genome stability and the DDR in the nervous system will have plenty of opportunities to keep themselves busy for the next several decades.

## Figures and Tables

**Figure 1 ijms-23-04142-f001:**
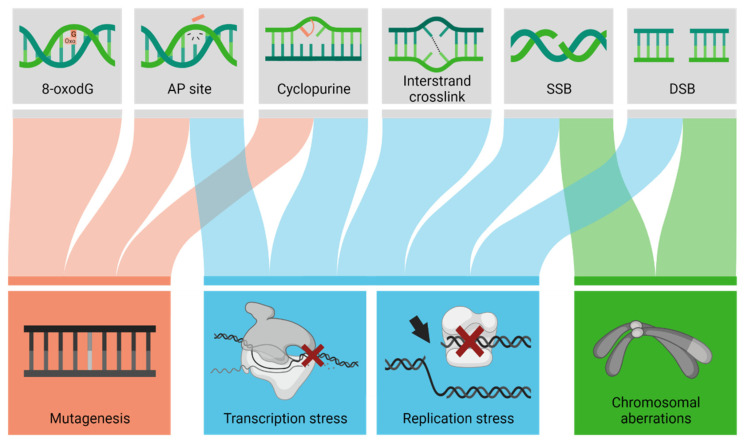
Unresolved DNA damage lesions can cause various molecular outcomes. Depending on the type of DNA lesion (top row), unresolved DNA damage can cause mutagenesis (change in the nucleotide sequence), transcription stress (arrest of an RNA polymerase), replication stress (collapse of the replication fork), chromosomal aberrations, or a combination of these outcomes (denoted by the colored lines). AP, apurinic/apyrimidinic; SSB, single-strand break; DSB, double-strand break.

**Figure 2 ijms-23-04142-f002:**
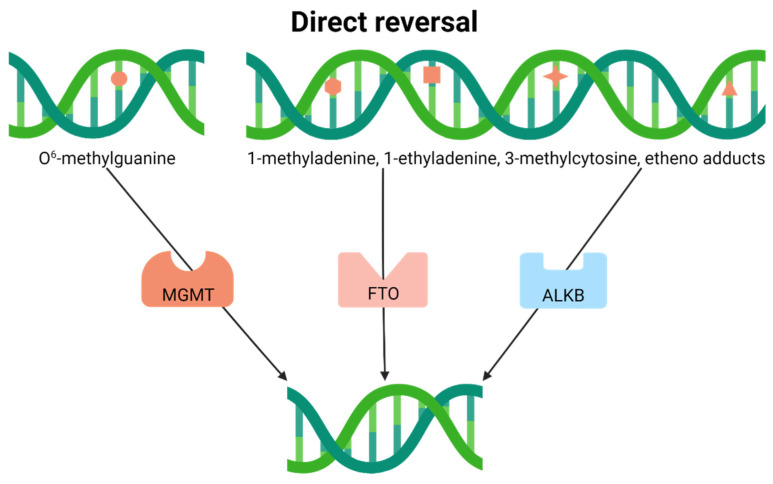
Direct reversal resolves subtle base modifications. The type of lesion determines the repair protein, which directly removes the modification without additional processing.

**Figure 3 ijms-23-04142-f003:**
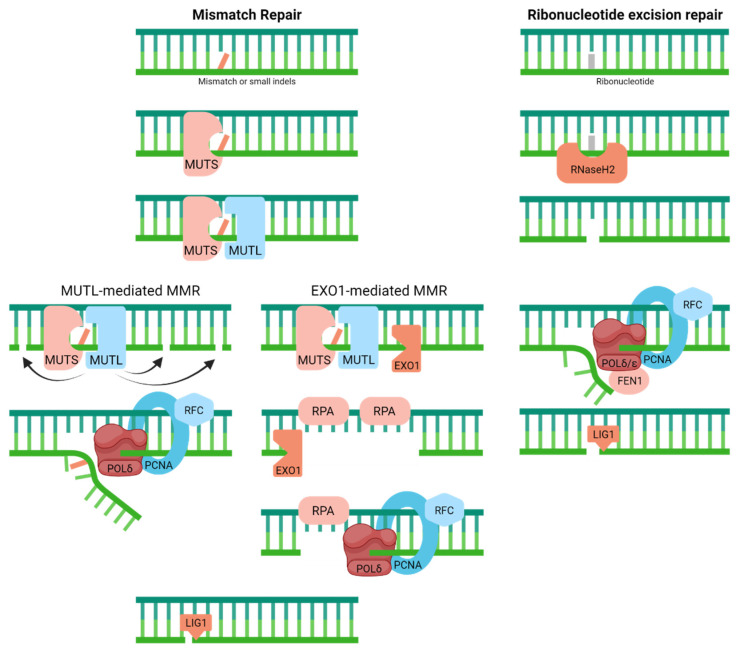
Mismatch repair resolves mispaired nucleotides or small insertions/deletions (indels), whereas ribonucleotide excision repair handles wrongly incorporated ribonucleotides within the DNA molecule. Full pathway descriptions and protein details can be found in the text and Table 1.

**Figure 4 ijms-23-04142-f004:**
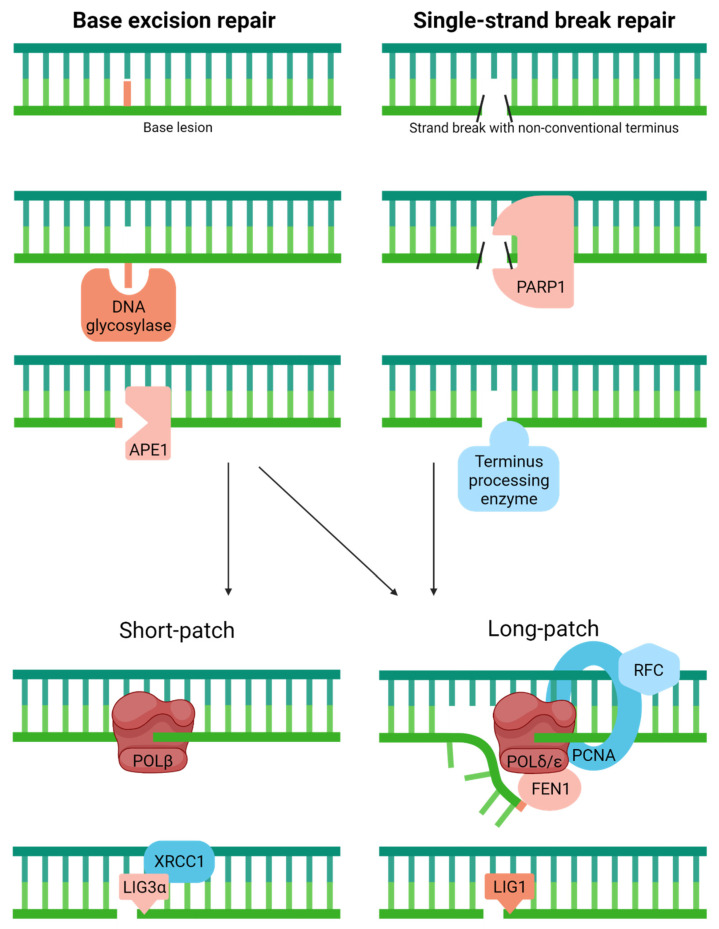
Classic base excision repair copes with various abnormal or modified bases, while single-strand break repair is a related, specialized pathway that resolves non-conventional 3′ or 5′ termini in DNA. Full pathway descriptions and protein details can be found in the text and Table 1. The large red line indicates the substrate base, whereas the smaller red box designates the 5′ sugar fragment prior to removal by POLꞵ (Short-patch) or during strand-displacement synthesis (Long-patch). The black slashes in SSBR represent various non-conventional ends processed by specific enzymes as discussed in the text.

**Figure 5 ijms-23-04142-f005:**
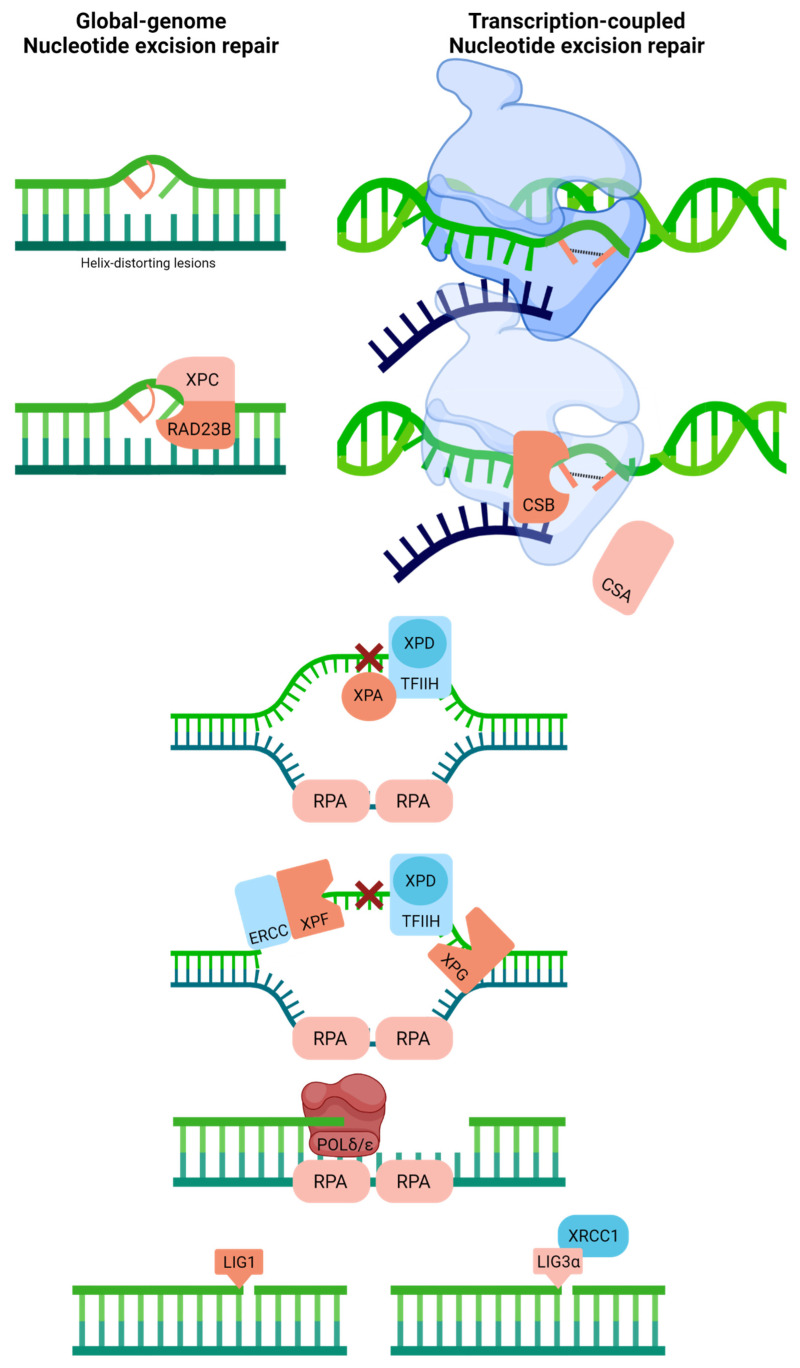
Nucleotide excision repair recognizes and repairs helix-distorting adducts like UV light-induced (6-4)photoproducts and cyclopyrimidine dimers. Two distinct pathways can be identified based on damage recognition, i.e., global-genome NER and transcription-coupled NER. Full pathway descriptions and protein details can be found in the text and Table 1. Red linked bases indicate UV photodimer substrate (or possibly another helix-distorting base damage). The blue strand designates synthesized RNA during transcription.

**Figure 6 ijms-23-04142-f006:**
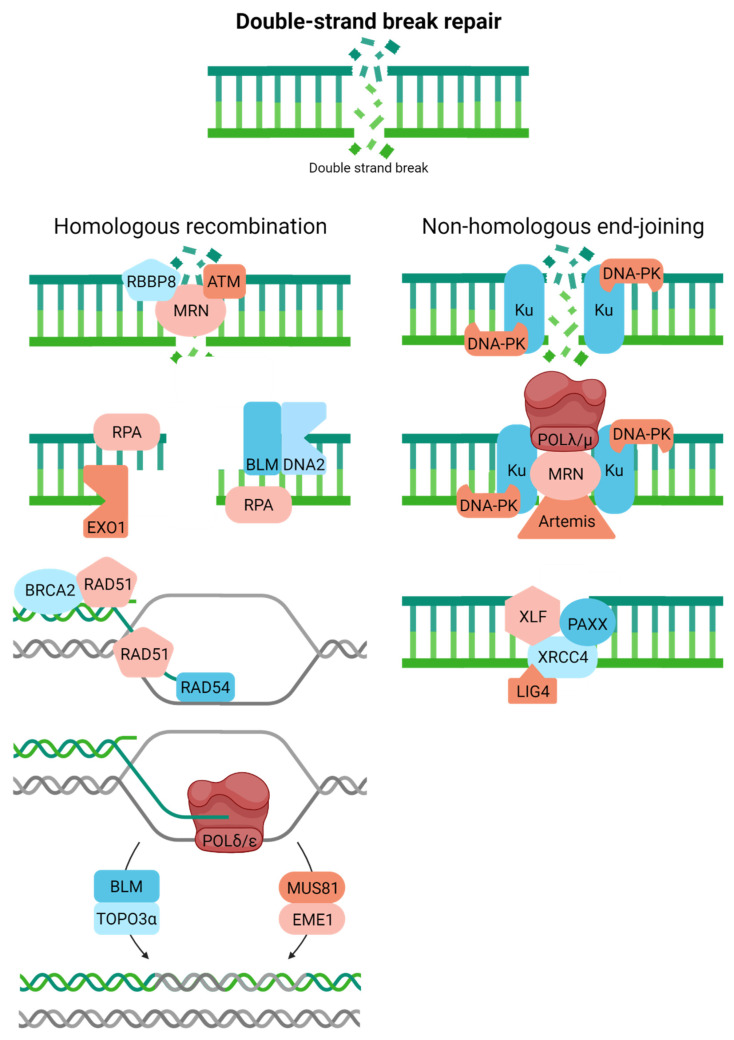
Double-strand breaks can be resolved via homologous recombination or non-homologous end-joining. Full pathway descriptions and protein details can be found in the text and Table 1. Homologous recombination is crudely drawn, with more extensive details available in [81].

**Figure 7 ijms-23-04142-f007:**
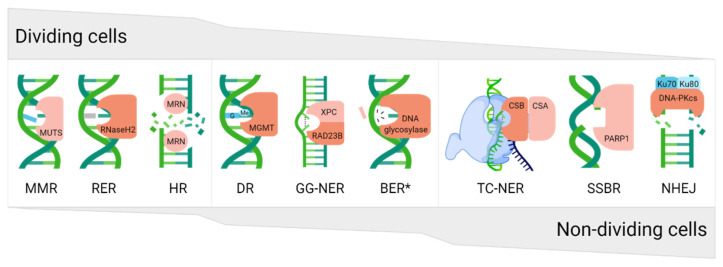
The proliferative character of cells allows them to preferentially use different DNA repair pathways. Dividing cells are more prone to use replication-associated pathways, e.g., mismatch repair (MMR), ribonucleotide excision repair (RER), or homologous recombination (HR), whereas non-dividing cells are more likely to apply non-homologous end-joining (NHEJ), single strand break repair (SSBR), or transcription-coupled nucleotide excision repair (TC-NER). Direct reversal (DR), global-genome NER (GG-NER), and base excision repair (BER) can be used in almost all cell types, regardless of proliferation status. The gray area indicates the level of pathway utilization in dividing (top) or non-dividing (bottom) cell types. For each pathway, only a representative protein is shown, typically the main recognition factor. * refers to general BER and not the long-patch sub-pathway, which engages several replication proteins and thus is relatively inactive in non-dividing cells.

**Table 1 ijms-23-04142-t001:** Core Factors of the Major DNA Damage Processing/Repair Mechanisms. For a more comprehensive list, see: https://www.mdanderson.org/documents/Labs/Wood-Laboratory/human-dna-repair-genes.html, accessed on 2 March 2022.

DNA Repair Gene	Full Name	Protein Biochemical Function	Genetic Disease/Disorder *	
**Modulation of Nucleotide Pools**				
DNPH1	2′-Deoxynucleoside 5′-Phosphate N-Hydrolase 1	Hydrolase for 5-hydroxymethyl deoxyuridine	NLE	
DUT	dUTP Pyrophosphatase	dUTPase	NLE	
NUDT1 (MTH1)	Nudix Hydrolase 1 (MutT Homolog 1)	8-oxoGTPase	NLE	
PARK7 (DJ1)	Park7 Gene (Oncogene DJ1)	Guanine glycation repair	Parkinson disease 7	●
**Direct Reversal**				
ALKB3 (DEPC1)	AlkB Homolog 3	1-meA dioxygenase	NLE	
ALKBH2 (ABH2)	AlkB Homolog 2	1-meA dioxygenase	NLE	
MGMT	Methylguanine-DNA Methyltransferase	O^6^-meG alkyltransferase	NLE	
**Mismatch Repair**				
EXO1	Exonuclease 1	5′ exonuclease	NLE	
MLH1	MutL Homolog 1	MutL homologs, forming heterodimer	Colorectal cancer, hereditary nonpolyposis, type 2 Mismatch repair cancer syndrome 1 Muir–Torre syndrome	
MSH2	MutS Homolog 2	Mismatch and loop recognition	Colorectal cancer, hereditary nonpolyposis, type 1 Mismatch repair cancer syndrome 2 Muir–Torre syndrome	
MSH3	MutS Homolog 3	Loop recognition	Endometrial carcinoma, somatic Familial adenomatous polyposis 4	
MSH6	MutS Homolog 6	Mismatch recognition	Endometrial carcinoma, familial Colorectal cancer, hereditary nonpolyposis, type 5 Mismatch repair cancer syndrome 3	
PMS2	PMS1 Homolog 2	MutL homologs, forming heterodimer	Colorectal cancer, hereditary nonpolyposis, type 4 Mismatch repair cancer syndrome 4	
**Base Excision Repair**				
APEX1 (APE1)	Apurinic Endonuclease 1	AP endonuclease	NLE	
MBD4	Methyl-CpG-Binding Domain Protein 4	DNA glycosylases: major altered base released: U or T opposite G at CpG sequences	NLE	
MPG (AAG)	N-Methylpurine DNA Glycosylase (3-Alkyladenine DNA Glycosylase)	DNA glycosylases: major altered base released: 3-meA, ethenoA, hypoxanthine	NLE	
MUTYH (MUTY)	MutY DNA Glycosylase	DNA glycosylases: major altered base released: A opposite 8-oxoG	Adenomas, multiple colorectal Gastric cancer, somatic	
NEIL1	Endonuclease VIII-Like 1	DNA glycosylases: major altered base released: Removes thymine glycol	NLE	
NEIL2	Endonuclease VIII-Like 2	DNA glycosylases: major altered base released: Removes oxidative products of pyrimidines	NLE	
NEIL3	Endonuclease VIII-Like 3	DNA glycosylases: major altered base released: Removes oxidative products of pyrimidines	NLE	
NTHL1 (NTH1)	Endonuclease III-Like 1	DNA glycosylases: major altered base released: Ring-saturated or fragmented pyrimidines	Familial adenomatous polyposis 3	
OGG1	8-Oxoguanine DNA Glycosylase	DNA glycosylases: major altered base released: 8-oxoG opposite C	Renal cell carcinoma, clear cell, somatic	
POLB	DNA Polymerase β	BER in nuclear DNA	NLE	
SMUG1	Single-Strand-Selective Monofunctional Uracil-DNA Glycosylase 1	DNA glycosylases: major altered base released: U	NLE	
TDG	Thymine-DNA Glycosylase	DNA glycosylases: major altered base released: U, T or ethenoC opposite G	NLE	
UNG	Uracil-DNA Glycosylase	DNA glycosylases: major altered base released: U	Immunodeficiency with hyper IgM, type 5	
**Ribonucleotide Excision Repair**				
RNASEH2	Ribonuclease H2	RNA-DNA ribonuclease	Aicardi–Goutieres syndrome 2	●
			Aicardi–Goutieres syndrome 3	●
			Aicardi–Goutieres syndrome 4	●
TOP1	DNA Topoisomerase I	Alter DNA topology by breaking single DNA strand	DNA topoisomerase U, camptothecin-resistant	
**Nucleotide Excision Repair**				
CETN2	Centrin 2	Binds DNA distortions	NLE	
DDB1	DNA Damage-Binding Protein 1	Complex defective in XP group E	NLE	
DDB2 (XPE)	DNA Damage-Binding Protein 2	Complex defective in XP group E	Xeroderma pigmentosum, group E, DDB-negative subtype	●
ERCC1	Excision Repair Complementing Defective in Chinese Hamster 1	5′ incision DNA binding subunit	Cerebrooculofacioskeletal syndrome 4	●
ERCC2 (XPD)	Excision Repair Complementing Defective in Chinese Hamster 2 (XPD Gene)	5′ to 3′ DNA helicase	Cerebrooculofacioskeletal syndrome 2 #	●
			Trichothiodystrophy 1, photosensitive	
			Xeroderma pigmentosum, group D	●
ERCC3 (XPB)	Excision Repair, Complementing Defective in Chinese Hamster 3 (XPB Gene)	3′ to 5′ DNA helicase	Trichothiodystrophy 2, photosensitive	
			Xeroderma pigmentosum, group B	●
ERCC4 (XPF)	Excision Repair Complementing Defective in Chinese Hamster 4 (XPF Gene)	5′ incision catalytic subunit	Fanconi anemia, complementation group Q	
			Xeroderma pigmentosum, group F	●
			Xeroderma pigmentosum, type F/Cockayne syndrome	●
			XFE progeroid syndrome	●
ERCC5 (XPG)	Excision Repair Complementing Defective in Chinese Hamster 5 (XPG Gene)	3′ incision	Cerebrooculofacioskeletal syndrome 3	●
			Xeroderma pigmentosum, group G	●
			Xeroderma pigmentosum, group G/Cockayne syndrome	●
ERCC6 (CSB)	Excision Repair Cross-Complementing Group 6	Cockayne syndrome and UV-Sensitive Syndrome; Needed for transcription-coupled NER	Susceptibility to lung cancer	
			Susceptibility to macular degeneration, age related, 5	
			Cerebrooculofacioskeletal syndrome 1	●
			Cockayne syndrome, type B	●
			De Sanctis–Cacchione syndrome	●
			Premature ovarian failure 11	
			UV-sensitive syndrome 1	
ERCC8 (CSA)	Excision Repair Cross-Complementing Group 8	Cockayne syndrome and UV-Sensitive Syndrome; Needed for transcription-coupled NER	Cockayne syndrome, type A	●
			UV-sensitive syndrome 2	
RAD23A	RAD23 Homolog A	Substitutes for RAD23B	NLE	
RAD23B	RAD 23 Homolog B	Binds DNA distortions	NLE	
UVSSA (KIAA1530)	UV-Stimulated Scaffold Protein A	Cockayne syndrome and UV-Sensitive Syndrome; Needed for transcription-coupled NER	UV-sensitive syndrome 3	
XPA	Xeroderma Pigmentosum Complementation Group A	Binds damaged DNA in preincision complex	Xeroderma pigmentosum, group A	●
XPC	Xeroderma Pigmentosum Complementation Group C	Binds DNA distortions	Xeroderma pigmentosum, group C	
**Strand Break Processing Factors and Helicases**				
APEX2	Apurinic/Apyrimidinic Endonuclease 2	AP endonuclease	NLE	
APLF	Aprataxin- and PNKP-Like Factor	Accessory factor for DNA end-joining	NLE	
APTX	Aprataxin	Processing of DNA single-strand interruptions	Ataxia, early-onset, with oculomotor apraxia and hypoalbuminemia	●
BLM	Bloom	Bloom syndrome helicase	Bloom syndrome	
LIG1	Ligase 1	DNA ligase	NLE	
LIG3	Ligase III	DNA Ligase III	NLE	
PNKP	Polynucleotide Kinase 3′ Phosphatase	Converts some DNA breaks to ligatable ends	Charcot-Marie-Tooth disease, type 2B2	●
			Ataxia-oculomotor apraxia 4	●
			Microcephaly, seizures, and developmental delay	●
RECQL (RECQ1)	RECQ Protein-Like	DNA helicase		
RECQL4	RECQ Protein-Like 4	DNA helicase	Baller-Gerold syndrome RAPADILINO syndrome Rothmund–Thomson syndrome, type 2	
RECQL5	RECQ Protein-Like 5	DNA helicase	NLE	
SPRTN (Spartan)	AprT-Like N-Terminal Domain Protein	Reads ubiquitylation	Ruijs–Aalfs syndrome	
TDP1	Tyrosyl-DNA Phosphodiesterase 1	Removes 3′-tyrosylphosphate and 3′-phosphoglycolate from DNA; human disorder SCAN1	Spinocerebellar ataxia, autosomal recessive, with axonal neuropathy 1	●
TDP2 (TTRAP)	Tyrosyl-DNA Phosphodiesterase 2 (TRAF- and TNF Receptor-Associated Protein)	5′- and 3′-tyrosyl DNA phosphodiesterase	Spinocerebellar ataxia, autosomal recessive 23	●
WRN	Werner	Werner syndrome helicase/3′–exonuclease	Werner syndrome	●
XRCC1	X-ray Repair Cross Complementing 1	Scaffold LIG3 accessory factor	Spinocerebellar ataxia, autosomal recessive 26 #	●
**Non-Homologous End-Joining**				
DCLRE1C (Artemis)	DNA Cross-Link Repair Protein 1C	Nuclease	Omenn syndrome Severe combined immunodeficiency, Athabascan type	
LIG4	Ligase IV	Ligase	Resistance to Multiple myeloma	
			LIG4 syndrome	●
NHEJ1 (XLF, Cernunnos)	Nonhomologous End-Joining Factor 1 (XRCC4-Like Factor)	End joining factor	Severe combined immunodeficiency with microcephaly, growth retardation, and sensitivity to ionizing radiation	●
PRKDC (DNA-PKcs)	Protein Kinase DNA-Activated Catalytic Subunit (DNA-Dependent Protein Kinase)	DNA-dependent protein kinase catalytic subunit	Immunodeficiency 26, with or without neurologic abnormalities	●
XRCC4	X-ray Repair Cross Complementing 4	Ligase accessory factor	Short stature, microcephaly, and endocrine dysfunction	●
XRCC5 (Ku80)	X-ray Repair Cross Complementing 5 (Ku Antigen, 80-KD Subunit)	DNA end binding subunit	NLE	
XRCC6 (Ku70)	X-ray Repair Cross Complementing 6 (Ku Antigen, 70-KD Subunit)	DNA end binding subunit	NLE	
**Homologous Recombination**				
BARD1	BRCA1-Associated Ring Domain 1	BRCA1-associated	Susceptibility to breast cancer	
BRCA1	Breast Cancer 1 Gene	Accessory factor for transcription and recombination, E3 Ubiquitin ligase	Breast-ovarian cancer, familial, 1 Susceptibility to pancreatic cancer, 4 Fanconi anemia, complementation group S	
EME1 (MMS4L)	Essential Meiotic Structure-Specific Endonuclease 1	Subunits of structure-specific DNA nuclease	NLE	
EME2	Essential Meiotic Structure-Specific Endonuclease 2	Subunits of structure-specific DNA nuclease	NLE	
GEN1	GEN1 Homolog of Drosophila	Nuclease cleaving Holliday junctions	NLE	
HELQ (HEL308)	Helicase PolQ-Like	DNA helicase in RAD51 paralog complex	NLE	
MRE11A	MRE11 Homolog	3′ exonuclease, defective in ATLD (ataxia-telangiectasia-like disorder)	Ataxia-telangiectasia-like disorder 1	●
MUS81	MUS81 Structure-Specific Endonuclease Subunit	Subunits of structure-specific DNA nuclease	NLE	
NBN (NBS1)	Nibrin	Mutated in Nijmegen breakage syndrome	Aplastic anemia	
			Leukemia, acute lymphoblastic	
			Nijmegen breakage syndrome	●
RAD50	RAD50 Double-Strand Break Repair Protein	ATPase in complex with MRE11A, NBS1	Nijmegen breakage syndrome-like disorder	●
RAD51	RAD51 Recombinase	Homologous pairing	Susceptibility to breast cancer	
			Fanconi anemia, complementation group R	
			Mirror movements 2	●
RAD51B	RAD51 Paralog B	RAD51 homolog	NLE	
RAD51D	RAD51 Paralog D	RAD51 homolog	Susceptibility to breast-ovarian cancer, familial, 4	
RAD52	RAD52 Homolog	Accessory factors for recombination	NLE	
RAD54B	RAD54 Homolog B	Accessory factors for recombination	Colon cancer, somatic Lymphoma, non-Hodgkin, somatic	
RAD54L	RAD54-Like	Accessory factors for recombination	Breast cancer, invasive ductal Adenocarcinoma, colonic, somatic Lymphoma, non-Hodgkin, somatic	
RBBP8 (CtIP)	Retinoblastoma-Binding Protein 8	Promotes DNA end resection	Jawad syndrome	●
			Pancreatic carcinoma, somatic	
			Seckel syndrome 2	●
SLX1A (GIYD1)	SLX1 Homolog A (GIY-YIG Domain Containing Protein 1)	Subunit of SLX1-SLX4 structure-specific nuclease, two identical tandem genes in the human genome	NLE	
SLX1B (GIYD2)	SLX1 Homolog B (GIY-YIG Domain Containing Protein 2)	Subunit of SLX1-SLX4 structure-specific nuclease, two identical tandem genes in the human genome	NLE	
SWI5	SWI5 Homologous Recombination Repair Protein	Accessory factor for loading RAD51	NLE	
XRCC2	X-ray Repair Cross Complementing 2	DNA break and crosslink repair	Fanconi anemia, complementation group U # Premature ovarian failure 17 # Spermatogenic failure	
XRCC3	X-ray Repair Cross Complementing 3	DNA break and crosslink repair	Susceptibility to breast cancer Melanoma, cutaneous malignant, 6	
**Fanconi Anemia Pathway**				
BRCA2 (FANCD1)	BRCA2 Gene (Fanconi Anemia, Complementation Group D1)	Cooperation with RAD51, essential function	Susceptibility to breast cancer, male	
			Breast-ovarian cancer, familial, 2	
			Glioblastoma 3	●
			Medulloblastoma	●
			Pancreatic cancer 2	
			Prostate cancer	
			Fanconi anemia, complementation group D1	
			Wilms tumor	
BRIP1 (FANCJ)	BRCA1-Interacting Protein 1 (Fanconi Anemia, Complementation Group J)	DNA helicase, BRCA1-interacting	Susceptibility to breast cancer, early-onset Fanconi anemia, complementation group J	
FAAP20	Fanconi Anemia-Associated Protein, 20-KD Subunit (Chromosome 1 Open Reading Frame 86)	Tolerance and repair of DNA crosslinks and other adducts in DNA: FANCA-associated	NLE	
FAAP24	FA Core Complex-Associated Protein 24	Tolerance and repair of DNA crosslinks and other adducts in DNA: FAAP24	NLE	
FAAP100	Fanconi Anemia-Associated Protein, 100-KD Subunit	Part of FA core complex	NLE	
FANCA	Fanconi Anemia, Complementation Group A	Tolerance and repair of DNA crosslinks and other adducts in DNA: FANCA	Fanconi anemia, complementation group A	
FANCB	Fanconi Anemia, Complementation Group B	Tolerance and repair of DNA crosslinks and other adducts in DNA: FANCB	Fanconi anemia, complementation group B	
FANCC	Fanconi Anemia, Complementation Group C	Tolerance and repair of DNA crosslinks and other adducts in DNA: FANCC	Fanconi anemia, complementation group C	
FANCD2	Fanconi Anemia, Complementation Group D2	Target for monoubiquitination	Fanconi anemia, complementation group D2	
FANCE	Fanconi Anemia, Complementation Group E	Tolerance and repair of DNA crosslinks and other adducts in DNA: FANCE	Fanconi anemia, complementation group E	
FANCG (XRCC9)	Fanconi Anemia, Complementation Group G (X-ray Repair Cross Complementing 9)	Tolerance and repair of DNA crosslinks and other adducts in DNA: FANCG	Fanconi anemia, complementation group G	
FANCI	Fanconi Anemia, Complementation Group I	Target for monoubiquitination	Fanconi anemia, complementation group I	
FANCL	Fanconi Anemia, Complementation Group L	Tolerance and repair of DNA crosslinks and other adducts in DNA: FANCL	Fanconi anemia, complementation group L	
FANCM	Fanconi Anemia, Complementation Group M	Helicase/translocase	Premature ovarian failure 15 # Spermatogenic failure 28	
PALB2 (FANCN)	Partner and Localizer of BRCA2 (Fanconi Anemia, Complementation Group N)	Co-localizes with BRCA2 (FANCD1)	Susceptibility to breast cancer Susceptibility to pancreatic cancer, 3 Fanconi anemia, complementation group N	
RAD51C (FANCO)	RAD51 Paralog C (Fanconi Anemia, Complementation Group O)	Rad51 homolog, FANCO	Susceptibility to breast-ovarian cancer, familial, 3 Fanconi anemia, complementation group O	
SLX4 (FANCP)	SLX4 Structure-Specific Endonuclease Subunit (Fanconi Anemia, Complementation Group P)	Nuclease subunit/scaffold SLX4, FANCP	Fanconi anemia, complementation group P	
UBE2T (FANCT)	Ubiquitin-Conjugating Enzyme E2T (Fanconi Anemia, Complementation Group T)	E2 ligase for FANCL	Fanconi anemia, complementation group T	
**DNA Damage Response Proteins**				
ATM	Ataxia-Telangiectasia Mutated Gene	Ataxia telangiectasia	Susceptibility to breast cancer	
			Ataxia-telangiectasia	●
			Lymphoma, B-cell non-Hodgkin, somatic	
			Lymphoma, mantle cell, somatic	
			T-cell prolymphocytic leukemia, somatic	
ATR	ATR Serine/Threonine Kinase	ATM- and PI-3K-like essential kinase	Cutaneous telangiectasia and cancer syndrome, familial #	
			Seckel syndrome 1	●
ATRIP	ATR-Interacting Protein	ATR-interacting protein	NLE	
CHEK1	Checkpoint Kinase 1	Effector kinases	NLE	
CHEK2	Checkpoint Kinase 2	Effector kinases	Susceptibility to breast and colorectal cancer Susceptibility to breast cancer Susceptibility to prostate cancer, familial Li-Fraumeni syndrome Osteosarcoma, somatic	
MDC1	Mediator of DNA Damage Checkpoint Protein 1	Mediator of DNA damage checkpoint	NLE	
PARP1 (ADPRT)	Poly(ADP-Ribose) Polymerase 1 (ADP-Ribosyltransferase 1)	Protects strand interruptions	NLE	
PARP2 (ADPRT2)	Poly(ADP-Ribose) Polymerase 2 (ADP-Ribosyltransferase 2)	PARP-like enzyme	NLE	
PARP3 (ADPRT3)	Poly(ADP-Ribose) Polymerase 3 (ADP-Ribosyltransferase 3)	PARP-like enzyme	NLE	
TP53	Tumor Protein 53	Regulation of the cell cycle	Adrenocortical carcinoma, pediatric	
			Basal cell carcinoma 7	
			Choroid plexus papilloma	
			Colorectal cancer	
			Glioma susceptibility 1	●
			Osteosarcoma	
			Bone marrow failure syndrome 5	
			Breast cancer, somatic	
			Hepatocellular carcinoma, somatic	
			Li-Fraumeni syndrome	
			Nasopharyngeal carcinoma, somatic	
			Pancreatic cancer, somatic	
TP53BP1 (53BP1)	Tumor Protein p53-Binding Protein 1	Chromatin-binding checkpoint protein	NLE	
**Accessory Proteins**				
HUS1	Hydroxyurea-Sensitive 1	Subunits of PCNA-like sensor of damaged DNA	NLE	
PCNA	Proliferating Cell Nuclear Antigen	Sliding clamp for pol delta and pol epsilon	Ataxia-telangiectasia-like disorder 2 #	●
RAD1	RAD1 Checkpoint DNA Exonuclease	Subunits of PCNA-like sensor of damaged DNA	NLE	
RAD9A	RAD9A Checkpoint Clamp Component A	Subunits of PCNA-like sensor of damaged DNA	NLE	
RAD17 (RAD24)	RAD17 Checkpoint Clamp Loader Component (Homolog of RAD24)	RFC-like DNA damage sensor	NLE	
RPA1	Replication Protein A1	Binds DNA in preincision complex	NLE	
RPA2	Replication Protein A2	Binds DNA in preincision complex	NLE	
RPA3	Replication Protein A3	Binds DNA in preincision complex	NLE	

* as specified in the OMIM database (https://omim.org/, accessed on 2 March 2022); # presumably indicates uncertain linkage; ● neurological symptoms/deficits; NLE, no link established.

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
