# Peer review of "Genome Integrity and Neurological Disease"

_ijms, 2022, doi:10.3390/ijms23084142_

Round 1

Reviewer 1 Report

In the review article entitled “Genome Integrity and Neurological Disease”, authors described cellular mechanisms useful to protect genomic integrity from insults coming from both external and internal sources. They also described how genetic mutations in DNA damage response genes lead to neuropathological manifestations. Authors indicate that this material serves as a prelude to the Special Issue, “Genome Stability and Neurological Disease”, where experts will discuss the role of DNA repair in preserving central nervous system function.  

The review is well written and organized; however, I think that a session aimed at introducing how changes in DNA damage response genes affect neurodevelopmental disorders is necessary. Indeed, they also mentioned that “the National Institute of Neurological Disorders and Stroke within the United States of America lists over 400 distinct neurological diseases on its website”, where neurogenetic diseases, developmental disorders, degenerative diseases, and trauma or infection-induced diseases are included.  While in the review authors properly anticipate the link between defective DNA damage responses and neurodegenerative disorders, they don’t mention at all the increasing studies in which social and cognitive defects occur concomitantly to defects in DNA repair proteins. For example, they should mention the important finding that learning and memory processes generate DNA double strand breaks (DSBs) in the promoters of a subset of early-response genes, including Fos, Npas4, and Egr1. Starting from these results it has been suggested that DSB formation is a physiological event that rapidly resolves topological constraints to early-response gene expression in neurons (Ram Madabhushi et al, Cell 2015). Other studies confirmed that the DNA repair associated protein Gadd45 regulates the temporal coding of immediate early gene expression required for the consolidation of associative fear memory. Thus, more and more findings are demonstrating that the correct brain function requires the proper DNA repair proteins machinery; for this reason, I suggest to add a session elucidating this point.

Author Response

Reviewer 1

Comment: In the review article entitled “Genome Integrity and Neurological Disease”, authors described cellular mechanisms useful to protect genomic integrity from insults coming from both external and internal sources. They also described how genetic mutations in DNA damage response genes lead to neuropathological manifestations. Authors indicate that this material serves as a prelude to the Special Issue, “Genome Stability and Neurological Disease”, where experts will discuss the role of DNA repair in preserving central nervous system function.  

The review is well written and organized; however, I think that a session aimed at introducing how changes in DNA damage response genes affect neurodevelopmental disorders is necessary. Indeed, they also mentioned that “the National Institute of Neurological Disorders and Stroke within the United States of America lists over 400 distinct neurological diseases on its website”, where neurogenetic diseases, developmental disorders, degenerative diseases, and trauma or infection-induced diseases are included.  While in the review authors properly anticipate the link between defective DNA damage responses and neurodegenerative disorders, they don’t mention at all the increasing studies in which social and cognitive defects occur concomitantly to defects in DNA repair proteins. For example, they should mention the important finding that learning and memory processes generate DNA double strand breaks (DSBs) in the promoters of a subset of early-response genes, including Fos, Npas4, and Egr1. Starting from these results it has been suggested that DSB formation is a physiological event that rapidly resolves topological constraints to early-response gene expression in neurons (Ram Madabhushi et al, Cell 2015). Other studies confirmed that the DNA repair associated protein Gadd45 regulates the temporal coding of immediate early gene expression required for the consolidation of associative fear memory. Thus, more and more findings are demonstrating that the correct brain function requires the proper DNA repair proteins machinery; for this reason, I suggest to add a session elucidating this point.

Response: We wish to thank the reviewer for this very insightful suggestion.  Indeed, within the DNA repair community, we often think of DNA damage per se as the only relevant feature that directs cell dysfunction and consequent death.  However, as the reviewer points out, emerging evidence indicates that DNA damage can be generated site-specifically in certain physiological contexts, and thus, efficient DNA repair may not only be needed to maintain genome integrity but to facilitate biological functional responses, such as memory consolidation.  As requested, we have added a new section on p. 11-12 (text in red) that highlights the recent studies by Madabhushi and Suberbielle and their colleagues, pointing out that DNA strand breaks are created and processed in genome-specific manners in the context of brain and neuronal functionality.  We believe that incorporation of this new material adds a unique element to our review article that will hopefully be recognized and appreciated by readers of the publication.

Reviewer 2 Report

Scheijen and Wilson well summarized the relationship between DNA stability and neurological diseases.

I think this review has value to publish in  IJMS, also hope to brush up on the article for wide-field readers.

Since so many events and factors are involved in DNA stability, a more illustrated description will be needed for general readers.

Especially, I confused the varieties of DNA repair responses against different DNA accidents. It is helpful to clarify what stress causes the distinct pathways and how each factor sequentially acts in DNA repair.

I also wonder how a variety of neurological diseases are caused by defects of distinct DNA repair pathways. Why do they show unique defects and what are common features? 

Author Response

Reviewer 2

Comment: Scheijen and Wilson well summarized the relationship between DNA stability and neurological diseases.

I think this review has value to publish in  IJMS, also hope to brush up on the article for wide-field readers.

Response: We thank the reviewer for their positive sentiment regarding our review article.

Comment: Since so many events and factors are involved in DNA stability, a more illustrated description will be needed for general readers.

Response: We initially left out the DNA repair pathway illustrations as these have been published in many formats in recent years.  However, in light of the concern raised, we have now included new figures that indicate the key molecular steps and protein participants in the major DNA repair pathways mentioned.  See new figures 2, 3, 4, 5 and 6.

Comment: Especially, I confused the varieties of DNA repair responses against different DNA accidents. It is helpful to clarify what stress causes the distinct pathways and how each factor sequentially acts in DNA repair.

Response: While we initially left out the pathway illustrations (see previous response), we had incorporated two figures that presented the repair pathway preferences based on replicative status (previous Figure 2, now 7) and the molecular consequences of specific types of DNA damage (Figure 1), with both figures eyeing potential outcomes relevant to the non-dividing nature of neurons.  Importantly, such illustrations have not previously been published, at least from what we could find.  However, the reviewer is correct that we have not identified the major substrates relevant to the different DNA repair mechanisms.  That information is now highlighted in the new pathway figures mentioned in the previous response.

Comment: I also wonder how a variety of neurological diseases are caused by defects of distinct DNA repair pathways. Why do they show unique defects and what are common features? 

Response: This is indeed a great question, and one, unfortunately, that we will not be able to answer.  As with cancer, it’s not fully clear why BRCA mutations give rise selectively to breast or ovarian cancers relative to other tissues/organs.  Our initial Figures were in a way trying to address the unique contributions of the different DNA repair mechanisms and the distinct effects of different types of DNA damage, with an eye on neurons (see previous response).  We have, in adding the new section noted in response to the concern of reviewer 1, that some brain region specificity might be dictated by certain DNA damage biological contexts, at least in terms of where that damage is generated and at what level.  Regardless, from our viewpoint, the topic of why certain brain regions are more susceptible to different types of DNA damage and thus defects in DNA repair remains a lingering issue in the field.